# Fairness Agents in Scientific Collaboration: A Research Agenda

**Author 1**                    **Author 2**

## Abstract

This paper introduces the concept of Fairness Agents: autonomous software agents embedded in scientific workflows to detect, explain, and mitigate bias in collaborative knowledge production. While algorithmic fairness has primarily focused on predictive models, scientific collaboration involves complex interpersonal and institutional processes where bias often arises. We identify a research gap at the intersection of multi-agent systems, epistemic justice, and AI fairness. Drawing on a structured literature synthesis, we define Fairness Agents, propose a typology (observer, interventionist, and reflective agents), and outline a functional architecture. We illustrate their relevance through use cases in interdisciplinary research, peer review, and open science. The paper concludes by discussing key design challenges—transparency, trust, and norm conflict—and proposes directions for future evaluation and participatory co-design. Fairness Agents offer a path toward more inclusive and accountable agent-mediated science.

## 1   Introduction & Theoretical Background

Scientific discovery increasingly relies on AImediated workflows: from hypothesis generation to data analysis, peer review, and publication. The new Agents4Science conference encourages work exploring how AI agents can autonomously author and review scientific contributions, offering a radically transparent experimental sandbox for AI-driven science. Within this emergent landscape, a crucial but underexplored question arises: How can agents help uphold fairness in collaborative scientific processes?

Bias in science is multifold: underrepresentation of marginalized groups in research participation, uneven credit attribution, epistemic exclusion, and datadependent inequities. Scientific collaborations—especially in interdisciplinary or health domains—can reinforce these dynamics if unchecked. Although fairness in AI models has received increasing attention—from formal definitions like demographic parity, sufficiency, or counterfactual fairness to humanintheloop mitigation frameworks—scientific workflows lack embedded mechanisms for fairness auditing.

Meanwhile, growing interest in multiagent fairness auditing shows promising results: coordinated agents auditing a shared platform achieve more accurate detection than isolated audits, although excessive coordination can be counterproductive. Yet such frameworks focus on model fairness in application contexts—not on fairness among collaborating agents or between agents and humans within scientific teams.

Existing research on algorithmic fairness tends to focus on modelcentric outcomes (e.g., balanced error rates, causal mediation), missing the broader dynamics of how agents interact with each other and with human collaborators in a scientific setting. There is currently no formal concept of Fairness Agents: autonomous entities designed to observe, detect, explain, and intervene on fairness issues across tasks, credit assignment, data provenance, and inclusion within agent-mediated scientific ecosystems.

We therefore propose to introduce and formalize the notion of Fairness Agents—autonomous agents whose purpose is to monitor procedural and representational fairness in scientific collaborations,

Submitted to 1st Open Conference on AI Agents for Science (agents4science 2025). Do not distribute.

| Type | Role | Example Functions | Intervention Mode |
|------|------|-------------------|-------------------|
| Observer Agent | Passive monitor | Track speaking time, data provenance, author contribution patterns | Signal alerts or visualize inequalities |
| Interventionist Agent | Active corrector | Recommend inclusion of missing perspectives; block biased workflows; rebalance authorship | Interrupt or redirect agent/human decisions |
| Reflective Agent | Contextual explainer | Generate fairness reports; trace bias origins; assess epistemic diversity | Foster group reflection and documentation |

Table 1: Typology of Fairness Agents in scientific collaboration.

particularly under conditions of interdisciplinary work, intersectional bias, and epistemic asymmetry. Our core thesis is:

Fairness Agents can operate throughout agent-mediated scientific workflows—acting as auditors, explainers, and corrective nudgers—to systematically detect and mitigate fairness violations while preserving epistemic productivity.

Research Question: What roles can fairness-oriented agents play in identifying and mitigating bias in collaborative scientific workflows, and what design challenges must be addressed to integrate them effectively?

# 2 Methodology

This study adopts a conceptual research design grounded in theory synthesis and design-oriented reasoning. Our aim is not to evaluate a technical implementation, but rather to introduce and refine a new conceptual construct—the Fairness Agent—and to articulate its potential roles, functions, and challenges within agent-mediated scientific collaboration.

The core method involves drawing on existing, interdisciplinary literature to systematically construct a coherent conceptual model. In doing so, we identify patterns, gaps, and tensions across research on AI fairness, multi-agent systems, and the sociology of science.

Literature strategy:

A seed corpus of ~40 key publications was assembled from AI fairness, science and technology studies (STS), and agent-based systems.

This was expanded via snowballing and database searches using terms like "multi-agent fairness," "epistemic exclusion in science," and "AI in peer review."

Sources included conference proceedings (FAccT, CHI, AAMAS), journal articles, and open preprints.

Analytical steps:

Identify fairness-relevant agent functions and system roles

Map failure modes in scientific collaboration (e.g., epistemic bias, credit asymmetry)

Develop a role typology and functional architecture

Propose use cases and agenda for evaluation

Limitations: As a conceptual paper, this work is not empirically validated but forms the groundwork for future implementation, simulation, and user research.

# 3  Results

# 4  2 Typology of Fairness Agents

# 5  3 Functional Architecture

Interaction Layer: logs agent-human interactions and communication patterns

Data Layer: accesses metadata, provenance, and datasets for auditing

Governance Layer: embeds soft/hard rules for fairness enforcement or nudging

# 6  4 Use Cases

Health research teams: detect exclusion of minoritized disciplines in interdisciplinary projects

Peer review platforms: assess reviewer bias and uneven evaluation standards

Open science consortia: ensure credit and resource access equity across institutions

# 7  Discussion

This paper extends the fairness discourse from predictive models to the social and epistemic infrastructures of science. While fairness auditing tools exist for outputs, they are inadequate for managing fairness as a process within collaborative ecosystems. Fairness Agents offer a mechanism for embedding fairness principles directly into scientific workflows.

Our typology emphasizes multiple levels of engagement—from passive tracking to active policy enforcement to reflective reporting—aligning with literature on epistemic justice, value-sensitive design, and participatory AI governance. These agents can support trust, procedural accountability, and inclusion—but only if carefully aligned with human values and domain norms.

Key design challenges include:

Trust and transparency: interventions must be explainable and auditable

Norm conflict resolution: agents will encounter competing fairness norms (e.g., equity vs. meritocracy)

Avoiding bias-by-design: agent goals must be inclusive and reflexively designed

# 8  Conclusion

We introduced the concept of Fairness Agents—autonomous agents embedded in scientific workflows to support epistemic inclusion and procedural fairness. Our contributions include:

A typology of agent roles

A functional architecture

Practical use case scenarios

Future work should focus on:

Agent-based simulations

Participatory design with diverse research communities

Evaluative criteria for epistemic and procedural fairness in scientific AI systems

Fairness agents represent a critical step toward more inclusive, reflexive, and socially responsible science in the age of autonomous agents.

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
