**Prompt Log for Submission: Fairness Agents in Scientific Collaboration: A Research Agenda?**

This document provides a transparent record of the prompts and instructions used in the collaborative writing of the conceptual paper submitted to the 'Agents for Science' conference, titled *Fairness Agents in Scientific Collaboration: A Research Agenda*. The log reflects the iterative process of human-AI collaboration.

**Initial User Prompt:**

*There is a conference for AI-generated papers. Here is the link to the conference: https://agents4science.stanford.edu/ Would you like to submit a paper?*

**User Selection of Topic:**

*I would suggest you go for Fairness Agents in with literature-based insights. You can start freely, I am here to supervise you if needed.*

**Literature Review/Theoretical Background**

*Start by providing Literature based on your theory and dive-deep in the research problem. Explain what you are doing, why you are doing it and what has already been done.*

You should propose a definition for Fairness Agent a typology of agent roles (e.g., observer, interventionist, reflective) with example functions, so that people can follow your space of interest.

**Methodology**

*You should explain more deeply what you found in the literature synthesis and why you chose this methodological approach. Generate example use cases.*

**Finalization**

*You should provide your manuscript with an abstract, summarizing your motivation, contribution, and key findings of the proposed Fairness Agent framework.*

**Critical Review & Limitations**

*You should list possible limitations of using large language models as co-authors in conceptual papers and Suggest ways to address hallucinated citations or shallow synthesis in AI-generated drafts.*

**7. Editing & Style**

*Ensure consistent academic tone and citation style (APA). Do check your submission for hallucinations and incorrect citations.*